# Comparison of *Campylobacter jejuni* Slaughterhouse and Surface-Water Isolates Indicates Better Adaptation of Slaughterhouse Isolates to the Chicken Host Environment

**DOI:** 10.3390/microorganisms8111693

**Published:** 2020-10-30

**Authors:** Katarina Šimunović, Sandra Zajkoska, Katja Bezek, Anja Klančnik, Darja Barlič Maganja, Sonja Smole Možina

**Affiliations:** 1Department of Food Science and Technology, Biotechnical Faculty, University of Ljubljana, Jamnikarjeva 101, 1000 Ljubljana, Slovenia; katarina.simunovic@bf.uni-lj.si (K.Š.); sandra.zajko@hotmail.com (S.Z.); anja.klancnik@bf.uni-lj.si (A.K.); 2Faculty of Health Sciences, University of Primorska, Polje 42, 6310 Izola, Slovenia; katja.bezek@fvz.upr.si (K.B.); darja.maganja@fvz.upr.si (D.B.M.)

**Keywords:** *Campylobacter jejuni*, slaughterhouse isolates, surface-water isolates, biofilm resistance, efflux pump activity

## Abstract

*Campylobacter jejuni* is an emerging food-borne pathogen that poses a high risk to human health. Knowledge of the strain source can contribute significantly to an understanding of this pathogen, and can lead to improved control measures in the food-processing industry. In this study, slaughterhouse and surface-water isolates of *C. jejuni* were characterized and compared in terms of their antimicrobial resistance profiles and adhesion to stainless steel and chicken skin. Resistance of *C. jejuni* biofilm cells to benzalkonium chloride and *Satureja montana* ethanolic extract was also tested. The data show that the slaughterhouse isolates are more resistant to ciprofloxacin, and adhere better to stainless steel at 42 °C, and at 37 °C in 50% chicken juice. Additionally, biofilm cells of the isolate with the greatest adhesion potential (*C. jejuni* S6) were harvested and tested for resistance to *S. montana* ethanolic extract, benzalkonium chloride, and erythromycin; and for efflux-pump activity, as compared to their planktonic cells. The biofilm cells showed increased resistance to both *S. montana* ethanolic extract and erythromycin, and increased efflux-pump activity. These data indicate adaptation of *C. jejuni* slaughterhouse isolates to the chicken host, as well as increased biofilm cell resistance due to increased efflux-pump activity.

## 1. Introduction

*Campylobacter* spp. infection and the resulting gastroenteritis represent >70% of all reported food-related human illnesses in the European Union (EU), which are mainly due to *Campylobacter jejuni* [1,2,3,4,5]. Although the major source of human campylobacteriosis is attributed to poultry, other implicated sources include ruminants, pets, and environmental sources, such as wildlife and water [6,7,8,9]. As shown before, chickens are the main source (60%) of *C. jejuni* that cause an invasive gastrointestinal illness with more severe clinical signs. Of the invasive strains, only 19% were attributed to environmental strains [10]. This makes environmental strains more ecologically interesting, while the comparison of strains according to source could provide valuable inside into *C. jejuni* behavior. Additionally, more antibiotic-resistant strains are expected in poultry compared to environmental sources [3]. This shows that strain characterization according to source can contribute significantly to prediction of severity of infection, which is therefore of crucial importance for human health.

Adhesion is the first step of biofilm formation, and it thus represents an important factor for survival and persistence of *C. jejuni* on abiotic surfaces, and contributes to the risk of cross-contamination [11,12]. In addition, bacterial adhesion and biofilm formation cause serious problems for hygiene and food spoilage, as well as increased corrosion rates of surfaces, which pose enormous problems for public health and the economy [13]. Biofilms and cells embedded in biofilms are also more resistant to antimicrobial agents, compared to planktonic cells. The physical barrier of the polymer matrix of the biofilm, the lower bacterial metabolism, the higher resistance to stressors, the efflux-pump activity, and the involvement of genetic elements all contribute to this biofilm resistance [14,15]. Stainless steel (SS) is commonly used in the food and beverage industry because it is corrosion-resistant, durable, and heat-resistant. However, organic material residues like food particles and chicken juice can act as a conditioning layer on SS surfaces, which can lead to increased bacterial adhesion [16,17,18,19]. Another source of microbial contamination in the poultry industry is chicken skin, which can contribute to microbial cross-contamination and significantly reduce the effectiveness of antimicrobial agents [17]. To reduce the burden of microbial contamination in food production, the use of effective cleaning and disinfecting agents is necessary, such as the quaternary ammonium compound benzalkonium chloride (BC) [20]. For the control of *Campylobacter* spp. in the food industry, the use of alternative antimicrobials should also be considered, such as antimicrobial plant extracts and plant-derived phenolic compounds, which can show better organoleptic properties and do not leave toxic residues [21].

The aim of the present study was to characterize *C. jejuni* isolated from a slaughterhouse and a surface-water environment according to (i) their individual antimicrobial resistance profiles; (ii) their adhesion to SS and to chicken skin; and (iii) the resistance of the biofilms formed on SS coupons against treatment with a plant extract (*Satureja montana* ethanolic extract; SMEE) and with BC; and to compare them according to their respective sources. Finally, biofilm cells of the isolate with the highest adhesion potential were harvested and tested for resistance to SMEE, BC, and erythromycin, and for the activity of their efflux pump, as compared to planktonic cells.

## 2. Materials and Methods

### 2.1. Bacterial Isolates and Growth Conditions

In the present study, *C. jejuni* isolated from slaughterhouse (*n* = 8) and surface-water environments (*n* = 8) were used (Table 1, Appendix A). These isolates were isolated and, to some extent, genetically and phenotypically characterized by Kovač et al. [8]. Sources, *flaA* nucleotide and peptide types, MLST clonal complex (CC) and sequence types (ST), mPCR clades, and year of isolation—obtained from Kovač et al. [8] are given in Appendix A. The isolates were obtained from a microbial collection from the Chair for Biotechnology, Microbiology and Food Safety, Biotechnical Faculty, University of Ljubljana (Ljubljana, Slovenia) and stored at −80 °C in Mueller–Hinton broth (MHB; Oxoid, UK) and glycerol (Kemika, Croatia) (80:20, *v*/*v*). Prior to the experiments, the isolates were grown on Mueller–Hinton agar (Oxoid, UK) at 42 °C under microaerobic conditions (5% O_2_, 10% CO_2_, 85% N_2_) for 24 h and replated one additional time. The replated culture was used in the experiments. For enumeration of *C. jejuni*, blood-free selective medium was used (Karmali medium; CM739; Oxoid).

### 2.2. Satureja Montana Ethanolic Extract 

For the extract preparation, 100 g dried *S. montana* herb (winter savory; Kottas Pharma, Austria) was added to 2-L 96% denatured ethanol (Roth, Germany) and mixed on a magnetic stirrer for 48 h at room temperature. The extract was then filtered using pleated paper filters (Rotilabo; Roth, Germany), and the filtrate was evaporated to dryness by rotary evaporation (175 bar; 40 °C; Rotavapor; Büchi, Switzerland). The final drying was carried out under a nitrogen flow, for a yield of 3 g dried extract from 100 g plant material. The dry SMEE was stored at −20 °C.

### 2.3. Antimicrobial Resistance

The susceptibilities of the individual *C. jejuni* isolates to six antimicrobials was tested—erythromycin, ciprofloxacin, tetracycline, gentamycin, nalidixic acid, and streptomycin—using commercially available plates (Sensititre EUCAMP2 plates; Thermo Fisher Scientific, UK), according to the manufacturer instructions. Briefly, the *C. jejuni* cultures of 5 × 10^5^ CFU/mL were prepared in MHB and added to the wells of the plates (100 µL/well). After 24 h incubation, the minimal inhibitory concentrations (MICs) were determined as the lowest concentration of the test antimicrobial that inhibited bacterial growth. The susceptibility of the individual isolates to each antimicrobial was evaluated according to the EUCAST [22] and Kahlmeter et al. [23] cut-off values. Each *C. jejuni* isolate underwent three biological replicates.

The resistance of the 16 individual *C. jejuni* isolates was also determined against SMEE and BC using the broth microdilution method, as for the MICs, and according to Kovač et al. [24]; similarly, for the resistance of the individual *C. jejuni* isolate S6 planktonic and biofilm cells to SMEE and BC, and also to erythromycin (see below). Briefly, SMEE was prepared at 2000 mg/L in MHB with 1% dimethylsulfoxide (DMSO), and BC was dissolved at 7.68 mg/L in MHB. Two-fold serial dilutions were made in flat-bottomed microtiter plates (Nunc, Thermo Scientific, Roskilde, Denmark). The *C. jejuni* cultures (at 5 × 10^5^ CFU/mL) were added to each well to a final volume of 100 μL (1:1, *v*/*v*). After 24 h, 10 µL resazurin solution (0.252 mg/mL resazurin, 0.14 mg/mL menadione, in MHB; Sigma Aldrich, Germany) was added to each well. After a 2-h incubation at 42 °C under microaerobic conditions, the plates were inspected visually to determine the MICs. The MICs were determined in triplicate.

### 2.4. Chicken Juice Preparation

Chicken juice was prepared as previously described [25]. Briefly, after freezing and thawing of chicken carcasses, the liquid was collected, filter sterilized (pore size, 0.45 μm; Sartorius, Göttingen, Germany), and stored at −20 °C.

### 2.5. Adhesion of Campylobacter jejuni

#### 2.5.1. Adhesion to Stainless Steel

For the SS adhesion assays, the suspension of each individual *C. jejuni* isolate was prepared in MHB to a final optical density (OD_600_) of 0.2. Electropolished American Iron and Steel Institute (AISI) 316 grade SS coupons (area, 1 cm^2^) that contained 18% chromium and 8% nickel were used [26]. Prior to use, the SS coupons were degreased in ethanol and sterilized by autoclaving at 121 °C for 15 min. One coupon was positioned in each well of a 48-well plate (Nunc, Thermo Scientific, Denmark) in an upright position. Then, 500 μL fresh MHB or MHB supplemented with chicken juice was added to each well, along with the prepared *C. jejuni* culture, to a final volume of 1 mL. The experimental conditions were as follows: (i) MHB medium at 42 °C under microaerobic conditions; (ii) MHB medium at 37 °C under microaerobic conditions; and (iii) MHB medium supplemented with 50% chicken juice at 37 °C under microaerobic conditions. Following a 72-h incubation, the *C. jejuni* suspensions were aspirated and the SS coupons were washed three times with phosphate-buffered saline (PBS; Oxoid, UK) to remove unattached *C. jejuni* cells. The coupons were then transferred into a fresh microtiter plate containing 1 mL PBS and treated in an ultrasonic bath (room temperature, 10 min; frequency, 28 kHz; power, 300 W; Iskra Pio, Šentjernej, Slovenia). They were then incubated on a shaker at 600 rpm (Thermo Fisher, UK) for 5 min, to release the adhered cells into the PBS. The number of adhered cells was determined by serial dilutions followed by the drop plate method and are presented as log numbers of colony forming units (log CFU/mL). *C. jejuni* adhesion to the SS coupons was determined in duplicate, with three independent experiments carried out for each isolate. The data gathered from testing each individual isolate was pooled to create two groups of isolates, namely, the slaughterhouse isolates and surface-water isolates.

#### 2.5.2. Adhesion to Chicken Skin

Adhesion of the slaughterhouse and surface-water *C. jejuni* isolates to chicken skin was determined as described previously [27,28], with some modifications. For the *C. jejuni* adhesion assay, chicken breast skin was removed from broiler carcasses and stored at −20 °C until use. Prior to the assay, the chicken skin was thawed for 1 h at room temperature, rinsed three times in sterile distilled water, aseptically cut (2 × 2 cm), and dried of excess liquid under a laminar flow. A piece of chicken skin was then placed into each well of sterile 12-well microtiter plates, with the addition of 100 µL *C. jejuni* suspension in PBS (1 × 10^8^ CFU/mL) to each well. A piece of skin incubated with sterile PBS treated in the same manner as the experimental samples was used as a negative control. After a 3 h incubation at 4 °C, each piece of chicken skin was transferred to a Falcon tube (50 mL; TPP, Switzerland) and rinsed three times with 3 mL PBS after short vortexing. In the final step, 1 mL of PBS was added to each piece of skin, with these samples sonicated in an ultrasonic bath (Elma Hans Schmidbauer, GmbH, Singen, Deutschland) for 10 min at 37 Hz. The number of *C. jejuni* cells that were released into suspension was determined after plating of serial dilutions on selective Karmali agar. The assays were performed in triplicate as three independent experiments per isolate. The data gathered from testing each individual isolate was pooled to create two groups of isolates, namely, the slaughterhouse isolates and surface-water isolates. The adhesion to the chicken skin of the *C. jejuni* isolates of different origins is presented as log CFU/mL.

### 2.6. Campylobacter jejuni Biofilm Treatment with SMEE and BC 

For biofilm formation, 1 mL cultures of the individual *C. jejuni* isolates at an OD_600_ of 0.1 was added to each well of 48-well microtiter plates with the SS coupons. To allow biofilm formation of *C. jejuni* on the SS coupons, the plates were incubated for 3 days at 42 °C under microaerobic conditions. For the biofilm treatments, SMEE was prepared in PBS with 1% DMSO to a final concentration of 1500 mg/L, and BC was prepared in PBS to a final concentration of 4.5 mg/L (5× mean MIC).

After biofilm formation, the SS coupons were washed three times with PBS, then treated with PBS, PBS with 1% DMSO (control), and SMEE or BC, for 1 min. After the treatment, the individual SS coupons were transferred into fresh microtiter plates. After three washes with PBS, the numbers of biofilm cells were determined as CFU/mL after 10-min treatment in an ultrasonic bath (room temperature, 10 min; frequency, 28 kHz; power, 300 W). The experiment was carried out three times for each tested isolate. The difference in the reduction of adhered cells on the SS coupons after the treatments are given, compared to the untreated control, as Δlog CFU/mL. The 1% concentration of DMSO had no adverse effects on *C. jejuni* viability.

### 2.7. Planktonic and Biofilm Cell Preparation and Their Antimicrobial Susceptibility

*Campylobacter jejuni* S6 planktonic cells were prepared from an initial suspension with OD_600_ 0.1 in MHB by culture dilution (in MHB), to match the concentration of cells obtained for the biofilm cell suspension. The numbers of planktonic cells are given as log CFU/mL.

To determine the resistance level and mechanism of biofilm resistance, *C. jejuni* S6 biofilm cells were prepared from the biofilms on the SS coupons after a 72 h incubation in MHB at 42 °C under microaerobic conditions, as previously described [29]. Briefly, after this incubation, the SS coupons were washed three times with PBS, and transferred into new microtiter plates containing 1 mL MHB. After sonication for 10 min, the cells released from the biofilms that were adhered to the SS coupons were transferred into the MHB. The numbers of biofilm cells are also presented as CFU/mL.

These planktonic and biofilm cells were incubated in MHB for 0, 4, 8, and 24 h at 42 °C under microaerobic conditions. At each time point, the numbers of the *C. jejuni* planktonic and biofilm cells were determined using serial dilutions, are expressed as CFU/mL, and their susceptibilities to erythromycin, SMEE, and BC were tested. These susceptibilities were determined according to the antimicrobial activity assays described above, at each of the sampling times (i.e., 0, 4, 8, 24 h), and are presented as MICs (mg/L). All of these assays were carried out in triplicate as three independent experiments.

### 2.8. Ethidium Bromide Accumulation 

The efflux pump activity of the planktonic and biofilm *C. jejuni* S6 cells was determined using ethidium bromide (EtBr) accumulation assays according to Kovač et al. [24], with some modifications. Briefly, 100 µL planktonic and biofilm cells were added to each well of black microtiter plates (Nunc, Thermo Scientific, Denmark), and then 3.125 µL of 16 mg/L EtBr (Sigma Aldrich, Germany) was added to each well, to give a final concentration of 0.5 mg/mL EtBr. Fluorescence measurements were carried out in a microplate reader (Varioskan Lux; Thermo Scientific, UK) using excitation at 500 nm and emission at 608 nm, at 37 °C for 1 h at 1 min intervals. The last 10 measurements were used for the EtBr accumulation calculations. The assays were carried out in triplicate, as three independent experiments. 

The efflux pump activity of *C. jejuni* S6 planktonic and biofilm cells was determined as the EtBr accumulation levels in relative fluorescent units (RFU) divided by the number of cells (colony forming units, CFU) in the 100 µL solution, for standardization. A higher value indicates lower activity of the efflux pumps, as more EtBr was accumulated in the cells.

### 2.9. Statistical Analysis

Mann–Whitney U tests were performed to test the differences in the antibiotic MIC levels between two *C. jejuni* isolate source groups. Chi-squared tests of independence were performed to examine the relationships between the isolate sources and the susceptibilities to the antibiotics (i.e., as resistant or sensitive). Independent sample *t*-tests were used to evaluate the differences according to (i) adhesion between slaughterhouse and surface-water isolates groups; (ii) biofilm reduction following treatments with SMEE and BC between the slaughterhouse and surface-water isolates groups; and (iii) EtBr accumulation between the planktonic and biofilm cells. All of the analyses were performed using the SPSS software, version 22 (IBM Corp., Armonk, NY, USA). The level of statistical significance was set at *p* < 0.05.

## 3. Results

### 3.1. Antibiotic Resistance Profiles of Campylobacter jejuni Slaughterhouse and Surface-Water Isolates 

In the present study, the susceptibilities of eight *C. jejuni* slaughterhouse isolates and eight surface-water isolates were tested individually for (i) the clinically important antibiotics erythromycin, ciprofloxacin, tetracycline, gentamicin, nalidixic acid, and streptomycin; (ii) SMEE, as a traditionally used herb; and (iii) BC, as a widely used disinfectant (Table 1 and Table 2).

All of the isolates tested were susceptible to erythromycin, gentamycin, and streptomycin. All of the slaughterhouse isolates were resistant to at least one of the antibiotics tested, compared to only three (of eight) resistant water isolates. For specific antibiotics, a higher prevalence of ciprofloxacin resistance was observed for the slaughterhouse isolates (88%; *n* = 7/8) compared to the surface-water isolates (25%; *n* = 2/8). In addition, a significantly higher probability of resistance of the slaughterhouse isolates to ciprofloxacin was observed (X2 = 6.349; *p* = 0.012). Furthermore, 50% (*n* = 4/8) of the slaughterhouse isolates were resistant to nalidixic acid, compared to only 25% (*n* = 2/8) of the surface-water isolates. This is similar for tetracycline, where 50% (*n* = 4/8) of the slaughterhouse isolates and 13% (*n* = 1/8) of the surface-water isolates showed resistance (Table 1).

Concurrent resistance to two antibiotics was seen for 88% (*n* = 7/8) of the slaughterhouse isolates and only 25% (*n* = 2/8) of the surface-water isolates. Of the slaughterhouse isolates, 50% (*n* = 4/8) were cross-resistant to ciprofloxacin and nalidixic acid, and 38% (*n* = 3/8) to ciprofloxacin and tetracycline. The cross-resistance of the surface-water isolates to ciprofloxacin and nalidixic acid was 25% (*n* = 2/8). Overall, no cross-resistance was found for the combination of ciprofloxacin, nalidixic acid, and tetracycline (Table 1).

The MICs for SMEE ranged from 61.5 mg/L to 500 mg/L; and for BC, from 0.24 mg/L to 1.92 mg/L. There was no link between SMEE and/or BC resistance and the origin of the isolates (Table 2).

### 3.2. Adhesion of Slaughterhouse and Surface-Water Campylobacter jejuni Isolates

The adhesion of the slaughterhouse and surface-water *C. jejuni* isolates to SS coupons was tested under different conditions: (i) in MHB medium at 37 °C and 42 °C; and (ii) in the presence of 50% chicken juice in MHB at 37 °C. To imitate the adhesion of *C. jejuni* in food processing, the adhesion to chicken skin was also tested as a biotic surface at 4 °C. The adhesion of *C. jejuni* isolates is shown as the number of cells in the biofilms adhering to the surfaces (i.e., SS coupons, chicken skin) after the 72 h (SS coupons) and 3 h (chicken skin) incubations (Figure 1).

The number of cells that adhered to the SS coupons in MHB under microaerobic conditions at 42 °C was significantly higher in the slaughterhouse isolates compared to the surface-water isolates (*p* = 0.029; Figure 1B), as well as for the SS coupons in MHB supplemented with 50% chicken juice under microaerobic conditions at 37 °C (*p* = 0.013; Figure 1C). There were no statistically significant differences in adhesion to SS coupons between these two groups when incubated in MHB under microaerobic conditions at 37 °C (Figure 1A), nor after inoculation and incubation of chicken skin in PBS at 4 °C (Figure 1D).

For the slaughterhouse isolates, the best adhesion to SS was in MHB supplemented with 50% chicken juice at 37 °C, followed by MHB at 42 °C and then at 37 °C. The adhesion of the surface-water isolates to SS did not significantly differ between the different conditions in MHB, although it appeared to be a little higher in MHB supplemented with chicken juice at 37 °C. These data indicate better adaptation of the slaughterhouse isolates to the chicken host environment (i.e., temperature of 42 °C and supplementation with chicken juice), compared to the surface-water isolates.

For the surface-water isolates, the adhesion to chicken skin was higher after only 3 h incubation at 4 °C, compared to the adhesion to SS coupons at 42 °C (Δ1.61 log units) and at 37 °C (Δ1.22 log units), and in MHB supplemented with 50% chicken juice at 37 °C (Δ0.3 log units). Thus, for the surface-water isolates, compared to the SS coupons, chicken skin appeared to be a more favorable surface for the adhesion of *C. jejuni*. Moreover, as well as showing an apparent higher adhesion rate compared to the SS coupons, the chicken skin showed less variation in this adhesion (SD = 0.21–0.20) compared to the SS coupons (SD = 0.32–1.20), for both of these isolate groups (Figure 1D).

### 3.3. Reduction of Campylobacter jejuni Biofilms on Stainless Steel by the Satureja montana Ethanolic Extract and Benzalkonium Chloride

The efficacies of SMEE as a plant-based alternative to commercial antimicrobials and BC as a widely used disinfectant were evaluated in terms of their promotion of *C. jejuni* biofilm removal from the SS coupons (Figure 2). The effects of SMEE were similar for both the slaughterhouse and the surface-water isolates, with mean reductions of 1.28 log units and 1.14 log units, respectively (*p* > 0.05). The biofilms of the slaughterhouse isolates on the SS coupons were, however, significantly more sensitive to the BC treatment compared to the surface-water isolates (*p* = 0.016), with mean reductions of adherent cells of 1.47 log units and 0.997 log units, respectively (Figure 2). This significantly lower reduction for the biofilms of the *C. jejuni* surface-water isolates compared to those of the slaughterhouse isolates indicate that they have a higher biofilm resistance to disinfectants such as BC, which was not seen for SMEE.

### 3.4. Resistance Profile of the Campylobacter jejuni S6 Biofilm Cells 

To determine whether the *C. jejuni* biofilm cells showed and/or maintained resistance after harvesting, the susceptibilities (i.e., MICs) of the *C. jejuni* S6 biofilm cells harvested from the SS coupon surfaces were examined after the further incubation periods of 0, 4, 8, and 24 h in MHB medium, with regard to erythromycin, SMEE and BC, and as compared to planktonic cells (Table 3). For erythromycin, two-fold higher MICs were observed for the biofilm cells compared to the planktonic cells immediately after their harvesting (0 h), which thus indicated greater resistance of the biofilm cells to erythromycin. Prolonged incubation of the planktonic and biofilm cells for up to 8 h showed an even greater difference in resistance to erythromycin. However, after 24 h of incubation, the MICs of the planktonic and biofilm cells were equal. In contrast, the resistance to SMEE of the biofilm cells compared to planktonic cells decreased over time, such that the MICs of the planktonic and biofilm cells were equal again after 24 h. Instead, the biofilm cells showed higher sensitivity than the planktonic cells to BC (fold-change = 0.5), which lasted up to the 8 h time point, although again, the MICs were equal after 24 h. These data indicate that the biofilm cells have greater resistance to the antibiotic but not to the disinfectant, and that these differences in biofilm cell susceptibility compared to those of the planktonic cells are lost over this prolonged (post-harvesting) incubation time.

### 3.5. Efflux Pump Activity of Planktonic and Biofilm Cells

The efflux pump activities of the *C. jejuni* S6 planktonic and biofilm cells were measured with the EtBr accumulation assay (Figure 3). After 4 h, the EtBr accumulation was significantly higher (3.28-fold; *p* = 0.002) in the biofilm cells compared to the planktonic cells, and even more so after 8 h (6.91-fold; *p* < 0.001). However, after 24 h, the levels of EtBr accumulation were similar for both cell types (1.10-fold difference). These data show that the changed state of the biofilm cells harvested from the biofilms on the SS coupons returns to the state of the planktonic cells after prolonged incubation time (i.e., 24 h).

## 4. Discussion

For almost a decade, C. *jejuni* has been the leading bacterial cause of foodborne diseases in humans. Due to the widespread overuse of antibiotics, the problem of antimicrobial resistance has increased and expanded worldwide [5,30,31]. The antibiotic resistance profiles of *C. jejuni* isolates from different environments can provide useful information for the development of strategies to regulate the use of antimicrobials in the food industry and in medicine. In the present study, the slaughterhouse isolates of *C. jejuni* showed higher antibiotic resistance, and better adaptation to the host and environmental conditions when compared to the surface-water isolates. In the case of ciprofloxacin, the prevalence of antimicrobial resistance of the slaughterhouse isolates was significantly higher compared to the surface-water isolates (Table 1). This ciprofloxacin resistance of *C. jejuni* slaughterhouse isolates (here as 88%) is consistent with Luo et al. [32] and Torralbo et al. [33], who showed that ciprofloxacin-resistant *Campylobacter* were present in 97% of isolates obtained from chickens. The report of the European Food Safety Authority and the European Centre for Disease Prevention and Control [5] that indicated 70% ciprofloxacin resistance in human isolates is therefore not surprising, as chickens are indeed the main source of *C. jejuni* infection [8,10,31]. According to the present study, 25% of the surface-water isolates were resistant to ciprofloxacin, which also indicates the importance of environmental sources of *C. jejuni* isolates that are resistant to clinically important antibiotics. However, this is a lower level compared to Szczepanska et al. [31], who reported ciprofloxacin resistance for 80% of *C. jejuni* isolates from dogs, and 85% resistance for pond isolates. Among their *C. jejuni* resistant isolates, 75% were resistant to ciprofloxacin and tetracycline [31]. This was also shown for 37.5% of all of the isolates in the present study. Although environmental sources appear to be responsible for only up to 10% of *C. jejuni* infections in humans, they still pose a clear threat to human infection in terms of resistant isolates [8,10,34]. As both poultry meat production and surface waters are potential sources of *Campylobacter* infection, the antimicrobial resistance of the isolates poses a major health risk, as it might lead to increasing numbers of difficult-to-treat cases of human campylobacteriosis [3].

Although there were differences in the antimicrobial resistance profiles between these slaughterhouse and surface-water *C. jejuni* isolates, there were no differences in the SMEE or BC susceptibilities, which showed comparable activities against these two groups. This confirms the observations of Kovač et al. [24], who showed that phenolic plant extracts are generally effective against both antibiotic-resistant and antibiotic-susceptible *Campylobacter* spp. regardless of the origin of the isolate. This thus indicates a different mode of action of these extracts, compared to the antibiotics.

Further differences between the slaughterhouse and surface-water *C. jejuni* isolates were observed for adhesion to SS coupons at 42 °C and in the presence of 50% chicken juice at 37 °C. As shown by Melo et al. [19], *C. jejuni* can establish an initial biofilm structure on polystyrene after only 4 h of incubation at 37 °C. Therefore, biofilm formation might be one of the factors that enable the survival and maintenance of *Campylobacter* outside the host, as was reported previously [19,35]. It has been shown that biofilm structures represent a cell reservoir, and that the presence of chicken juice significantly increases biofilm formation of *C. jejuni* isolates on inert surfaces in the food industry [18,19]. The present study suggests that isolates that show greater adhesion to abiotic surfaces when exposed to conditions closer to the chicken host (i.e., 42 °C; with chicken juice) have been selected under the conditions of the slaughterhouses and food processing. Similar data were shown by Sulaeman et al. [36], where isolates isolated from a food processing environment showed better adhesion to polystyrene at 42 °C than to live animals or carcasses.

Most *C. jejuni* cells are located in the crevices and feather follicles of the chicken skin, where they can remain even after water rinsing. The spiral-shaped *C. jejuni* cells can even penetrate the skin [37,38], which might explain the low adhesion variability of the slaughterhouse and surface-water *C. jejuni* isolates in the present study. The adhesion of all of the tested *C. jejuni* isolates to the chicken skin after 3 h at refrigerator temperature was comparable. Therefore, the high adhesion of all of these isolates to the chicken skin confirmed in the present study represents a risk for transmission of *C. jejuni* through the food chain regardless of the origin of the isolates.

As biofilms are a health risk in the food industry, their removal is a crucial step to ensure food safety. Although the adhesion and biofilm formation of bacteria provide us with crucial information, how susceptible these biofilms are to various antimicrobial treatments remains an open question. These data are of great importance for the correct performance of cleaning processes in food processing environments. Therefore, we compared the resistance of *C. jejuni* from established biofilms to SMEE and BC treatment. Although BC had lesser effects on the surface-water isolates compared to the slaughterhouse isolates, there were no differences between these two groups after the treatment with SMEE. Additionally, while the prevalence of antibiotics resistance and SS adhesion was higher in the slaughterhouse isolates, their biofilm cells were more susceptible to BC treatment. This indicates greater adaptation to harsh environments outside the host and increased survival rates of the surface-water isolates, which are advantages in harsh environments. In addition, biocides might pose a problem in the context of food safety and public health. Alternatively, several bioactive compounds of natural origin have shown antibiofilm activities against *Campylobacter* spp., such as resveratrol, *trans*-cinnamaldehyde, eugenol, carvacrol, and even olive oil by-products [39,40]. Indeed, according to the present study, SMEE might represent an alternative for the removal of adherent *C. jejuni* cells.

Biofilms are known for their higher resistance to antimicrobials and adverse environmental conditions. This resistance might be a consequence of reduced penetration of these substances into the biofilm itself, and/or of the development of more resistant cell forms within biofilms [14,15]. Interestingly, there were dynamic changes in the sensitivities of these *C. jejuni* biofilm cells to erythromycin, SMEE, and BC over different incubation periods compared to the planktonic cells. The increased resistance of these biofilm cells to erythromycin and SMEE is consistent with previous studies for several microorganisms, including *Staphylococcus aureus*, *Pseudomonas aeruginosa*, *Escherichia coli*, *Pseudomonas pseudomallei*, *Streptococcus sanguis*, and C. jejuni [41,42,43]. Biofilm cells are phenotypically different from planktonic cells [43], and as shown in the present study, they have a two-fold increased susceptibility to BC. This suggests that *C. jejuni* isolates have different resistance mechanisms to BC compared to erythromycin and SMEE.

The mechanism of increased resistance of biofilm cells was confirmed here in terms of their increased efflux pump activities. Efflux pumps have been described as an important mechanism in the resistance of *C. jejuni* against various antibiotics, disinfectants, and phenolic antimicrobials [21,24,44,45]. In addition, the involvement of the CmeABC and CmeDEF efflux pumps in the resistance of biofilm cells to the natural compound resveratrol has been demonstrated [29]. Interestingly, both the changes in the resistance of these biofilm cells against erythromycin, SMEE, and BC and their efflux pump activities reverted to the original planktonic cell activities after 24 h of (postharvest) incubation in MHB.

## 5. Conclusions

In the present study, significant differences in various characteristics of *C. jejuni* were shown with respect to the origins of the isolates, in terms of their antimicrobial resistance and adhesion to a SS surface. These data indicate greater adaptation of the slaughterhouse isolates to the chicken host environment, compared to the surface-water isolates as a group, regardless of some phenotypic variabilities within groups. This observation should also be confirmed for a larger sample size. Furthermore, the higher resistance of the *C. jejuni* biofilm cells shown here can be attributed to both the physical barrier of the biofilm matrix and the higher efflux pump activity of the biofilm cells. However, this resistance was reversible, and it thus needs to be investigated further in more detail.

## Figures and Tables

**Figure 1 microorganisms-08-01693-f001:**
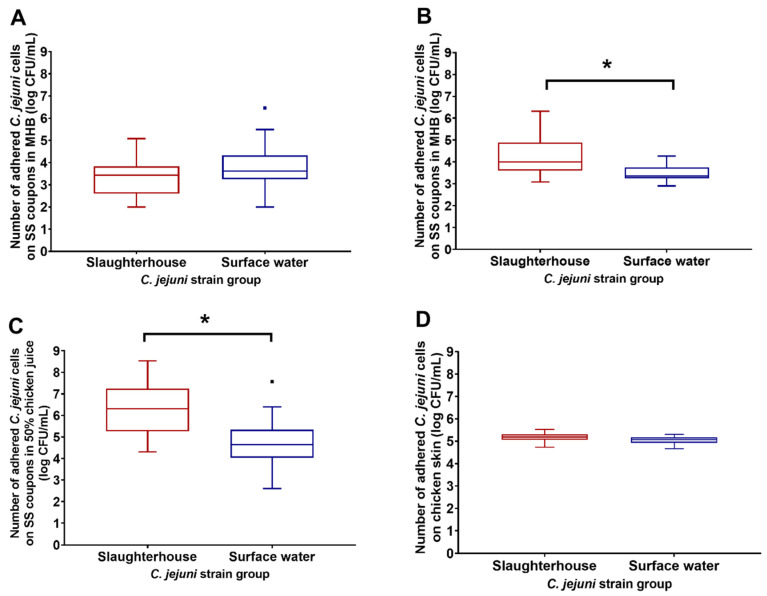
Adhesion (log CFU/mL) as the pooled data for the *Campylobacter jejuni* slaughterhouse and surface-water environment isolates to stainless steel (SS) coupons in Mueller–Hinton broth (MHB) for 72 h under microaerobic conditions at 37 °C (**A**) and 42 °C (**B**), in MHB with 50% chicken juice for 72 h under microaerobic conditions at 37 °C (**C**), and to chicken skin in phosphate-buffered saline (PBS) for 3 h at 4 °C (**D**). CFU—colony forming units. * Statistical significance at *p* < 0.05; dots present outliers.

**Figure 2 microorganisms-08-01693-f002:**
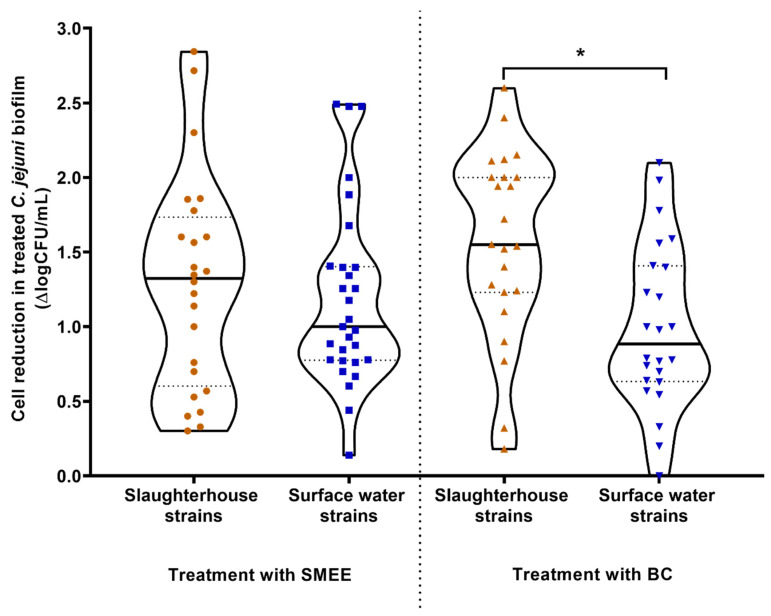
Effects of 1-min treatments with *Satureja montana* ethanolic extract (SMEE; left) and benzalkonium chloride (BC; right) as the pooled data for the biofilms of the *Campylobacter jejuni* slaughterhouse and surface-water environment isolates for their changes in adhesion to the stainless steel coupons. Continuous lines, medians; dotted lines, quartiles; * *p* < 0.05 (Student’s *t*-test).

**Figure 3 microorganisms-08-01693-f003:**
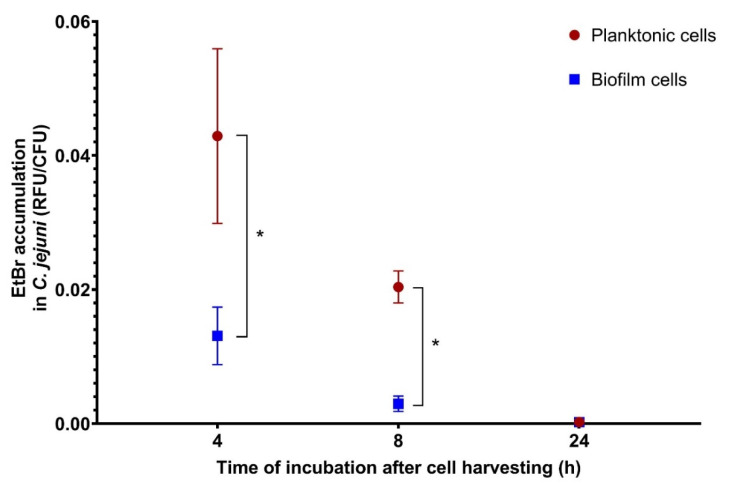
Ethidium bromide accumulation in *Campylobacter jejuni* S6 planktonic and biofilm cells. RFU, relative fluorescent units; CFU, colony forming units. * *p* < 0.05 (Student’s *t*-test).

**Table 1 microorganisms-08-01693-t001:** Susceptibilities of the individual *Campylobacter jejuni* slaughterhouse and surface-water isolates to the tested antibiotics.

Isolate Designation	Erythromycin		Ciprofloxacin		Tetracycline		Gentamicin		Nalidixic Acid		Streptomycin	
	MIC (mg/L)	S/R	MIC (mg/L)	S/R	MIC (mg/L)	S/R	MIC (mg/L)	S/R	MIC (mg/L)	S/R	MIC (mg/L)	S/R
Slaughterhouse												
S1	2	S	16	R	<0.5	S	0.25	S	64	R	0.5	S
S2	<1	S	0.25	S	4	R	0.25	S	2	S	1	S
S3	<1	S	8	R	8	R	<0.12	S	8	S	0.5	S
S4	<1	S	8	R	0.5	S	0.5	S	32	R	1	S
S5	<1	S	8	R	4	R	0.25	S	1	S	0.5	S
S6	<1	S	8	R	8	R	0.25	S	1	S	1	S
S7	<1	S	8	R	<0.5	S	0.5	S	64	R	1	S
S8	<1	S	8	R	<0.5	S	0.25	S	32	R	0.5	S
Surface water												
W1	<1	S	<0.25	S	<0.5	S	0.25	S	4	S	1	S
W2	<1	S	4	S	16	R	0.25	S	16	S	1	S
W3	<1	S	8	R	<0.5	S	0.25	S	64	R	1	S
W4	<1	S	<0.12	S	<0.5	S	0.5	S	4	S	1	S
W5	<1	S	0.12	S	<0.5	S	0.25	S	8	S	<1	S
W6	<1	S	0.12	S	<0.5	S	0.5	S	8	S	<1	S
W7	<1	S	16	R	0.5	S	0.5	S	>64	R	2	S
W8	<1	S	<0.12	S	<0.5	S	0.5	S	4	S	1	S

S/R, sensitive or resistant isolate, according to the clinical resistance cut-offs of European Committee on Antimicrobial Susceptibility Testing [22]. MIC—minimal inhibitory concentration.

**Table 2 microorganisms-08-01693-t002:** Susceptibilities of the individual *Campylobacter jejuni* slaughterhouse and surface-water isolates to the *Satureja montana* ethanolic extract and benzalkonium chloride.

Main Source	Isolate	MIC (mg/L)	
		*S. montana* Ethanolic Extract	Benzalkonium Chloride
Slaughterhouse	S1	250	0.96
	S2	250	0.96
	S3	500	0.96
	S4	250	0.24
	S5	250	0.48
	S6	500	0.96
	S7	125	0.48
	S8	250	1.92
Surface water	W1	500	0.96
	W2	250	0.96
	W3	250	0.48
	W4	125	0.96
	W5	250	0.96
	W6	250	0.96
	W7	500	1.92
	W8	61.5	0.96

**Table 3 microorganisms-08-01693-t003:** Resistance of *Campylobacter jejuni* S6 planktonic and biofilm cells against erythromycin, *Satureja montana* ethanolic extract and benzalkonium chloride with time of post-harvesting incubation in MHB.

Incubation Time (h)	MIC (mg/L)									
	Erythromycin			*S. montana* Ethanolic Extract			Benzalkonium Chloride		
	P	B	FC	P	B	FC	P	B	FC
0	0.125	0.25	2	125	250	2	0.48	0.24	0.5
4	0.125	0.25	2	250	250	1	0.48	0.24	0.5
8	0.125	0.5	4	250	125	0.5	0.48	0.24	0.5
24	0.125	0.125	1	500	500	1	0.96	0.96	1

P—planktonic cells; B—biofilm cells; FC—fold change.

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
