# Peer review of "Comparison of Campylobacter jejuni Slaughterhouse and Surface-Water Isolates Indicates Better Adaptation of Slaughterhouse Isolates to the Chicken Host Environment"

_microorganisms, 2020, doi:10.3390/microorganisms8111693_

Round 1

Reviewer 1 Report

Microorganisms-981286 is a resubmission of microorganisms-771180 which is dealing with the properties of some Campylobacter isolates from slaughterhouse and surface water. These properties include antibiotic resistance, biofilm formation, efflux-pump activity and resistance to antimicrobial agents like benzalkonium chloride, a plant extract and erythromycin. The screening of different strains of Campylobacter is of importance due to its high presence in food and food related environments and the food-related human illness.

The authors took under consideration the comments of the editors and reviewers, and re-submitted an improved manuscript.  The text is well written and minor changes are required. Minor spell check is required by the authors e.g.

L375 to ensure

L380 to SMEE and BC treatment.

L382 groups after the treatment with SMEE.

Author Response

Response to reviewers

We are thankful for the acceptance of the manuscript to Microorganisms and are glad to further improve the manuscript with provided corrections. Below, please find answers to reviewers comments and described changes in the manuscript.

Reviewer 1

Microorganisms-981286 is a resubmission of microorganisms-771180 which is dealing with the properties of some Campylobacter isolates from slaughterhouse and surface water. These properties include antibiotic resistance, biofilm formation, efflux-pump activity and resistance to antimicrobial agents like benzalkonium chloride, a plant extract and erythromycin. The screening of different strains of Campylobacter is of importance due to its high presence in food and food related environments and the food-related human illness.

The authors took under consideration the comments of the editors and reviewers, and re-submitted an improved manuscript.  The text is well written and minor changes are required. Minor spell check is required by the authors e.g.

L375 to ensure

L380 to SMEE and BC treatment.

L382 groups after the treatment with SMEE.

Answer 1. Thank you for your observations. Changes have been made as follows:

L376 from ‘ensuring’ to ‘ensure’

L381 from ‘treatments with SMEE and BC’ to ‘SMEE and BC treatment’

L383 from ‘for the treatment’ to ‘after the treatment’

L41 a comma (,) added

L67 ‘and’ removed

L107 and L108 ‘against’ changed into ‘to’

L196 ‘of’ was added

Reviewer 2 Report

The paper offers significant points of interest.

More guidance should be given regarding the use of frozen skin samples.

It would be important to clarify why.

Author Response

Response to reviewers

We are thankful for the acceptance of the manuscript to Microorganisms and are glad to further improve the manuscript with provided corrections. Below, please find answers to reviewers comments and described changes in the manuscript.

Reviewer 2

The paper offers significant points of interest.

More guidance should be given regarding the use of frozen skin samples.

It would be important to clarify why.

Answer 2. To clarify the method, additional clarification has been added in L145 ‘until use’ and L146 ‘thawed for 1 hour at room temperature’. Chicken skin was frozen for storage and freezing has an additional benefit of reducing naturally occurring microbiota. Nevertheless, frozen skin was not used in experiments. Skin samples were thawed prior to use.

This manuscript is a resubmission of an earlier submission. The following is a list of the peer review reports and author responses from that submission.

Round 1

Reviewer 1 Report

The manuscript “Comparison of Campylobacter jejuni slaughterhouse and surface-water strains indicates adaptation of slaughterhouse strains to the chicken host environment” describe differences in various characteristics of C. jejuni with respect to the origins of the strains, in terms of their antimicrobial resistance and adhesion to a SS surface. These data indicate greater adaptation of the slaughterhouse strains to the chicken host environment, compared to the surface‐water strains.

The authors mentioned in the conclusion that the observation should be confirmed for a larger sample size. I do agree with the authors that a larger sample size is needed, the authors are strongly recommended to study significantly more strains and thereafter rewrite the manuscript.

All strains sampled at slaughterhouse level were sampled from different sampling sites. In this kind of study more than one strain should be sampled from each sampling site.

Can the authors ensure that the eight strains are eight different strains? The strain number B964, B977 and B979 have either the same MIC values, or only differ with one titer in the MIC value of a few different antimicrobial substances, which is within the margin of error for the analysis. As the sample B964, B977 and B979 are numbers close to each other and collected from, cecum, chicken skin after cooling and worker gloves after slaughter it might have been sampled at the same day and at the same slaughterhouse. The authors are recommended to ensure that it is not the same strain sampled at three different sites at slaughter but originated from the same flock. The same question applies regarding B972 and B975. The authors should whole genome sequence all the isolates to ensure that one strain is only included once in the study. This should also strengthen the study to compare the association between sequence types and antimicrobial resistance.

Reviewer 2 Report

The current manuscript is dealing with the properties of some Campylobacter isolates from slaughterhouse and surface water. These properties include antibiotic resistance, biofilm formation, efflux-pump activity and resistance to antimicrobial agents like benzalkonium chloride, a plant extract and erythromycin. The screening of different strains of Campylobacter is of importance due to its high presence in food and food related environments and the food-related human illness. Some parts of this manuscript are well written and descriptive whereas other parts are hard to understand and important information are missing. Please refer below for specific comment per section

Title:  the title is confusing as the reader may misunderstand that only the slaughterhouse isolates are capable of forming biofilm on chicken host environment; please revise

Introduction: L41-44 the take home message is not clear. Please clarify the meaning of these sentences; transitional words may help the reader to understand what the authors want to point out.

Materials and methods:

section 2.1: this section need an extended revision. It is not clear if the isolates were recovered from this study; if not please cite the previous study. It is also unclear which method used to characterize the isolates what is the meaning of some extent. The method used to characterize the isolates is of importance. It seems that the authors confuse and misuse the term “strain” instead of term “isolate”. Which method used to discriminate the isolates to strain level? if the isolates were not characterized at strain level please replace the term “strain” with “isolate” throughout the text including title, abstract and tables/figures.

L80: passage? Please use a more appropriate word for this

L84: “for the extraction…” which extraction?

Results:

In the case that the isolates characterization is a part of this study, please provide the obtained data

Section 3.2-3.3 I am not able to understand why in this part the adhesion variability of Campylobacter isolates is masked under the terms of the source of isolation. It will be of interest to have also the information for each specific isolate in order to be able to compare them with the rest properties tested

Conclusion: the importance of strain variability is hided in this section

Minor

L220-221: bold fond

Table3 the words are cut; please revise accordingly